# Neural correlates of perceiving and interpreting engraved prehistoric patterns as human production: Effect of archaeological expertise

Mathilde Salagnon[1], Sandrine Cremona[1], Marc Joliot[1], Francesco d'Errico[2,3ᵒ], Emmanuel Mellet[1ᵒ]*

**1** CNRS, CEA, IMN, GIN, UMR 5293, Université Bordeaux, Bordeaux, France, **2** PACEA UMR 5199, CNRS, Université Bordeaux, Pessac, France, **3** SFF Centre for Early Sapiens Behaviour (SapienCE), University of Bergen, Bergen, Norway

ᵒ These authors contributed equally to this work.
* emmanuel.mellet@u-bordeaux.fr

## Abstract

It has been suggested that engraved abstract patterns dating from the Middle and Lower Palaeolithic served as means of representation and communication. Identifying the brain regions involved in visual processing of these engravings can provide insights into their function. In this study, brain activity was measured during perception of the earliest known Palaeolithic engraved patterns and compared to natural patterns mimicking human-made engravings. Participants were asked to categorise marks as being intentionally made by humans or due to natural processes (e.g. erosion, root etching). To simulate the putative familiarity of our ancestors with the marks, the responses of expert archaeologists and control participants were compared, allowing characterisation of the effect of previous knowledge on both behaviour and brain activity in perception of the marks. Besides a set of regions common to both groups and involved in visual analysis and decision-making, the experts exhibited greater activity in the inferior part of the lateral occipital cortex, ventral occipitotemporal cortex, and medial thalamic regions. These results are consistent with those reported in visual expertise studies, and confirm the importance of the integrative visual areas in the perception of the earliest abstract engravings. The attribution of a natural rather than human origin to the marks elicited greater activity in the salience network in both groups, reflecting the uncertainty and ambiguity in the perception of, and decision-making for, natural patterns. The activation of the salience network might also be related to the process at work in the attribution of an intention to the marks. The primary visual area was not specifically involved in the visual processing of engravings, which argued against its central role in the emergence of engraving production.

**Data Availability Statement:** Raw Bold values and behavioural data are available at DOI 10.17605/OSF.IO/UQD79.

**Funding:** EM & FD: CNRS project 80 Prime Neurobeads, a grant from the IdEx Bordeaux/CNRS (PEPS 2015). FD: European Research Council through a Synergy Grant for the project Evolution of Cognitive Tools for Quantification (QUANTA), No. 951388; the Research Council of Norway through its Centres of Excellence funding scheme, SFF Centre for Early Sapiens Behaviour (SapienCE), project number 262618, the Talents Programme the Bordeaux University [grant number: 191022_001] and the Grand Programme de Recherche 'Human Past' of the Initiative d'Excellence (IdEx) of the Bordeaux University. The funders had no role in study design, data collection and analysis, decision to publish, or preparation of the manuscript.

**Competing interests:** The authors have declared that no competing interests exist.

# Introduction

The cognitive abilities of our prehistoric ancestors and how they evolved have become a crucial area of research in archaeology and anthropology [1–4]. Different research strategies are followed to investigate this topic. Past cognition can be inferred by analysing the material culture prehistoric populations have left behind, under the assumption that behavioural patterns reflect cognitive processes. A wide range of past behaviours have been investigated in this perspective, such as subsistence strategies [5, 6], stone and bone tool-making [7–15], containers [16], pigments [17–21], tool hafting [22, 23], mortuary practices [24, 25], ornamental objects [26–28], engraving and painting of cave walls and objects [29, 30]. More recently, past cognition has become the subject of interdisciplinary research combining archaeological data with methods and concepts from neuroscience [31–33].

Neuroarchaeology, as it has been termed, aims to create conceptual frameworks for modelling the evolution of human cognition in light of advances in the neurosciences, and to test such models experimentally based on data collected from modern participants. Research in this domain has investigated the potential co-evolution of tool-making and language by studying the overlap of the brain networks mobilised by these two skills [34–38]. The implication of executive functions and working memory in the production of knapped stone tools, involving different levels of cognitive control and neural substrates depending on the complexity of the practised stone tool technology, has also been the subject of studies [34, 35, 39, 40].

The emergence of symbolic behaviour has also been investigated recently by neuroarchaeology. Some archaeologists have argued that the earliest graphic manifestations, dating from the Lower and Middle Palaeolithic in Eurasia and the African Middle Stone Age, were conceived and used as signs or symbols, and thus demonstrate abstraction and communication capacities that were not previously attributed to the human populations of those times [41–49]. Others contend that early abstract engraving production resulted from low-level visual perceptual phenomena [50–52] and should be interpreted as a "proto-aesthetic" behaviour devoid of semiotic intent. Still others see the production of abstract engravings as resulting from kinaesthetic dynamics of a non-representational sort that allowed hominins to engage and discover the semiotic affordances of mark-making [53], or as decorative, cultural transmitted patterns with no apparent symbolic meaning [54]. In a previous study [55], we characterised the neural basis of the visual processing of prehistoric abstract engravings dated between 540,000 and 30,000 years before the present, and showed that despite their relatively simple structure, engraving perception engaged the visual cortices of the ventral visual pathway that are involved in the recognition and identification of objects.

Consistent with the view of their being representational in nature, our first results showed that the primary visual area was not sensitive to the global organisation of the engravings, and thus did not support the previously suggested hypothesis that this region played a specific and exclusive role in the emergence and perception of the production of early engravings [50, 56]. The debate stimulated by these findings [57, 58] and, in particular, the criticism that inferences drawn from experiences with present-day humans could be inadequate for understanding perceptual processes specific to our prehistoric ancestors, makes it necessary to develop strategies to overcome this potential drawback to the extent possible.

Attributing intentional human agency to abstract marks is a prerequisite for using them as a medium for culturally-mediated indexical communication. Our ancestors needed to distinguish purposely made engravings from other accidental or natural marks in order to recognise their communicative potential and use them as means to store, transmit and retrieve meaning. It is reasonable to assume that if abstract engravings were used as signs or symbols by our ancestors, the latter must have shared a knowledge that allowed them to recognise the

engravings as the result of a conscious, deliberate, technical action intended to embody meaning in a tangible medium. In our previous study [55], the participants lacked archaeological knowledge. The brain regions mobilised by the perception of the engravings be altered according to the level of familiarity that the subjects have with these productions. The inclusion of participants with this familiarity allows approaching the knowledge that the engravers probably possessed and thus avoid a novelty effect at the brain level in the participants [59, 60]. To simulate this knowledge, we included archaeologist participants who are familiar with or experts in prehistoric engravings. We compared them at both behavioural and brain functional levels to a control group with no such expertise, paired for age, gender, and level of education. The first aim of the present work was to estimate the effect of familiarity and prior knowledge, hereafter referred to as *Expertise*, on the brain regions involved in the perception of abstract engravings and their attribution to human agency. The present study investigated this effect in a "Judgment" task where participants had to assess whether past humans had intentionally produced the marks on objects, or whether the marks resulted from natural processes such as erosion, carnivore gnawing or root etching. Therefore, this study explored whether familiarity modifies the regions involved in the visual processing of engravings, particularly in the primary visual area. The second aim of the study was to assess whether the attribution of the marks to human *versus* non-human agency could be differentiated at the functional brain level, and to what extent such difference could be conditioned by the observer's expertise.

## Materials and methods

### Participants

Thirty-one healthy adults with no neurological history were included after providing written informed consent to participate in the study. They were divided into two groups according to their expertise in Palaeolithic archaeology: Controls, without any prior background in the discipline (n = 15, mean age ± SD: 44 ± 10 years, range: 30–63 years, six women, none left-handed) and Experts, i.e. scholars actively working in the discipline with knowledge in Palaeolithic art and bone modifications (n = 16, mean age ± SD: 44.6 ± 10 years, range: 32–61 years, six women, one left-handed). The two groups of participants were matched for age, gender, and education level (PhD, 20 years of schooling after first grade).

### Ethics statements

The 'Sud-Ouest outremer III' local Ethics Committee approved the study (N° = 2016-A01007-44).

### MRI acquisition

The blood oxygen level-dependent (BOLD) signal was mapped in the 31 volunteers using functional magnetic resonance imaging (fMRI) with a Siemens Prisma 3 Tesla MRI scanner. The structural images were acquired with a high-resolution 3D T1-weighted sequence (TR = 2000 ms, TE = 2.03 ms; flip angle = 8°; 192 slices and 1 mm isotropic voxel size). The functional images were acquired with a whole-brain T2*-weighted echo-planar image acquisition (T2*-EPI Multiband x6, sequence parameters: TR = 850 ms; TE = 35 ms; flip angle = 56°; 66 axial slices and 2.4 x 2.4 x 2.4 mm isotropic voxel size). The functional images were acquired in three runs during a single session. The experimental design was programmed using E-prime software (Psychology Software Tools, Pittsburgh, PA, USA). The stimuli were displayed on a 27" screen. The participants viewed the stimuli through the magnet bore's rear via a mirror mounted on the head coil.

## Description of the task

Participants performed a judgment task based on the visual presentation of pictures of intentionally human-made and natural marks. The judgment task included two conditions: Attribution ("is the mark intentionally made by a human being?") or Orientation ("is the longest axis of the medium on which the marks are present vertical?"). The orientation task is a control condition. It used the same images as the attribution condition in a task that does not require visual analysis of the marks (defining the orientation of the longest axis of the object without paying attention to the marks present on them) nor any archaeological knowledge. When subtracting the activations of the orientation task from those of the attribution task, all the activations that are not specific to the latter (low-level perceptual processes such as contrast, luminance, perception of the shape of the support, motor activity related to button press. . .) are cancelled-out. For each stimulus, the type of judgment to be made (i.e. Attribution or Orientation) was displayed during 0.5s, before the stimulus was presented. Then the stimulus was presented for 3s (Fig 1). Participants had to answer "yes" or "no" by clicking on a response box as soon as the stimulus was replaced by the one-second reminder of the instruction ("human?" or "vertical?"). During the baseline, a fixation cross was displayed and a square appeared after a variable delay (3.5s ± 1s). Participants had to click on the response box as soon as the square appeared (Fig 1). The participants saw a total of 21 different human-made marks and 21 different natural marks divided into three runs lasting 5 min and 57 sec each, presented in a randomized order. Participants thus saw the item twice, once in the Attribution judgement and once in the Orientation judgement.

**Stimuli.** The 21 pictures of engravings included in the study were abstract engravings, dated between 800 ka to 30 ka, not found in Upper Paleolithic contexts in association with figurative art, have demonstrated anthropogenic origin [29, 61, 62, see S1 Table], and were recognizable on a photo of the object on which they occur. The number of items (21 human and 21 non-human) was chosen in order not to tire the participants since we adopted a so-called slow event-related paradigm (a 3s presentation every 9.5s). The engravings come from African and Eurasian sites, and are attributed to *Homo erectus*, Neanderthals and Early Modern Humans.

The original pictures were converted into greyscale and put on a grey background (Fig 2, left). The natural marks category included 21 objects in different materials bearing modifications produced by natural modelling of the bone surface (e.g. imprints of nerves and vascular canals), gnawing by carnivores, root etching, erosion, and fossilisation of plants [63]. Pictures were converted into greyscale and displayed on a grey background (Fig 2, right).

**Post fMRI session debriefing.** After the fMRI session, the participants were asked to indicate the criteria on which they had based their decision. The criteria were: shape of the marks, criss-cross patterns, presence of parallel marks, repetition of identical marks, depth of the marks, number of marks and the nature of medium of the marks.

In addition, the experts were asked whether they had ever seen any of the engravings.

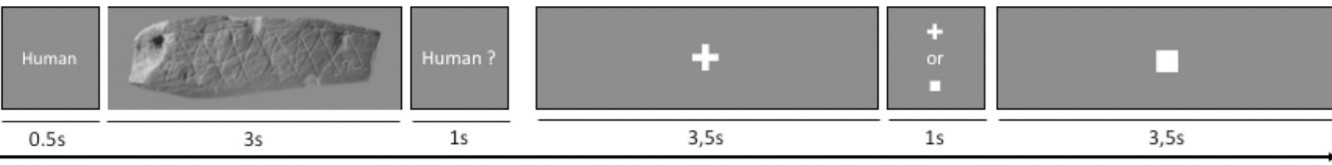

**Fig 1. Organization of a trial in the judgment task.** Participants were presented each item twice (once during the Attribution and once during the Orientation task). The participants were shown 21 different human-made and 21 natural marks.

Human                                                Non Human

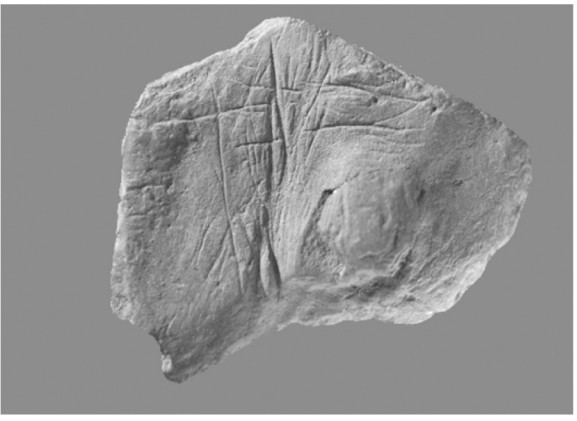 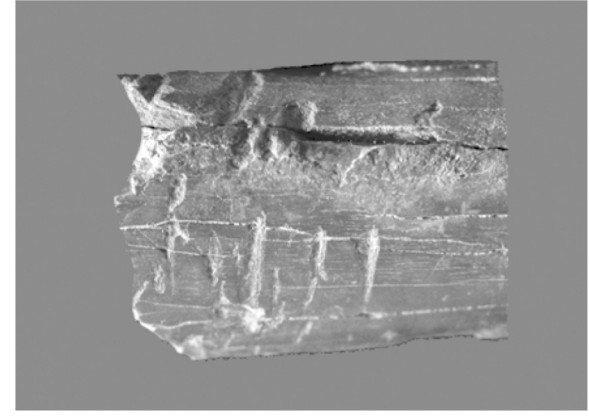

**Fig 2. Examples of stimuli used in the judgment task.** Left: human stimulus (engraving from Blombos Cave, Southern Africa, c. 77,000 years old). Right: non-human marks due to carnivore gnawing.

## Data analysis

**Preprocessing.** Functional volumes were processed using Nipype, which allows the different steps to be chained together [64]. The T1-weighted scans of the participants were normalised to a site-specific template, matching the MNI space using the SPM12 'segment' procedure with the default parameters. To correct for subject motion during the fMRI runs, the 192 EPI-BOLD scans were realigned within each run using a rigid-body registration. Then, the EPI-BOLD scans were rigidly registered structurally to the T1-weighted scan. The combination of all the registration matrices allowed warping of the EPI-BOLD functional scans to the standard space with trilinear interpolation. Once in the standard space, a 5 mm FWHM Gaussian filter was applied.

**First level analysis.** For each subject, global linear modelling (GLM, statistical parametric mapping (SPM 12), http://www.fil.ion.ucl.ac.uk/spm/) was used for processing the task-related fMRI data, with effects of interest (tasks) being modelled by boxcar functions corresponding to paradigm timing, convolved with the standard SPM hemodynamic temporal response function. We then computed the effect of interest-related individual contrast maps, corresponding to each experimental condition. Note that 8 regressors of no-interest were included in the GLM analysis: time series for WM, CSF (average time series of voxels belonging to each tissue class), the six motion parameters and the temporal linear trend.

**Analysis of behavioural response.** To assess whether the observed correct response rates were different from chance, we calculated the 95% confidence interval of a random response rate for 42 trials. Rates outside the 34–66% range were considered significantly different from chance.

To estimate the effect of Expertise on correct response rates, we analysed the behavioural responses for Attribution and Orientation separately, since the distribution of the correct response rate for the Orientation condition was not Gaussian. We used a non-parametric Wilcoxon test to evaluate performance differences between Experts and Controls in the Orientation condition.

To test whether the effect of Expertise depended on the type of judgment made in the Attribution condition, we estimated the interaction effect between Expertise and Attribution on the correct response rate, using a linear mixed-effect model fitting random effects at the participant level. A two-way interaction term between Expertise and Attribution (and their lower-

order terms) was set as the fixed effect predictors, and correct response rate as the dependent variable. The significance of fixed effects was assessed through ANOVA components.

**Analysis of debriefing data.** To assess the effect of Expertise on the criteria used to discriminate intentional human marks versus non-human ones, we computed a chi-squared test for each of the seven criteria.

**Analysis of fMRI data.** Group analysis of fMRI data was carried out using JMP®, Version 15. SAS Institute Inc., Cary, NC, 1989–2019. A first step was to select the regions that were activated significantly in the contrast of interest, namely [Attribution *minus* Orientation]. We extracted signal values from the first-level analysis maps of each of the 192 homotopic regions of interest (hROI) of the AICHA functional atlas [65] for each experimental condition. Two hROIs were excluded from the analysis because of a lack of signal in at least 15% of their volume: gyrus_parahippocampal-4 (19% non-signal) and Thalamus-8 (46.66% non-signal). The hROIs included in the analysis fulfilled two criteria in each group of participants: 1. Significantly more activated in the [Attribution *minus* baseline (cross fixation)] contrast (univariate t-test p < 0.05 uncorrected) to discard deactivated hROIs. 2. Significantly more activated in the [Attribution *minus* Orientation] contrast (univariate t-test p < 0.05 FDR corrected) to discard activation not specific to Attribution. hROIs selected for Experts and Controls were grouped to obtain the final list of hROIs included in the subsequent analysis.

To assess the effect of Expertise on BOLD activations according to the Attribution response (human or non-human marks), a mixed-effect linear regression model was implemented on the BOLD values of the 64 hROIs activated in the [Attribution minus Orientation] contrast. A three-way interaction term between hROIs (64) X Expertise (Experts, Controls) X Attribution (Human, Non-human) and all lower order terms was set as the fixed effect predictors, BOLD values as the dependent variable and random effects were fitted at the participant level. The significance of fixed effects was assessed through ANOVA components.

## Results

### Behavioural results

In the Attribution condition, Experts gave 81.3% (mean) ± 15% (SD) of correct responses (for both human and non-human attribution) while Controls responded correctly to 61.3% (mean) ± 17% (SD) of the items. The number of correct responses in Orientation did not differ between Experts and Controls (88.1% ± 14% and 86.7% ± 17% respectively, p = 0.96, Wilcoxon), thus showing, as expected, that the expertise effect was present in Attribution but not in Orientation condition.

We did not observe any significant interaction between Expertise and Attribution ($F_{(1,29)}$ = 0.56, p = .46, Fig 3). However, the linear mixed-effect model revealed a main effect of Expertise, with Experts exhibiting better performances than Controls ($F_{(1,29)}$ = 31.3, p< 0.0001), and a main effect of Attribution, as the rate of correct responses was higher for human than non-human judgments ($F_{(1,29)} =$ 14.3, p< 0.0007). Thus, whatever the type of judgment made, experts had a better rate of correct response than controls on average and, whatever the level of expertise, the correct response rate was higher on average for human than non-human judgment.

### Debriefing results

The decision criteria reported by the participants for attributing a human agency to abstract marks were repetition of identical marks, shape of the marks, presence of parallel marks, and presence of criss-cross patterns. Note that the engravings of European origin are mainly made of parallel in pattern, whereas African engravings often show cross-patterns However, none of

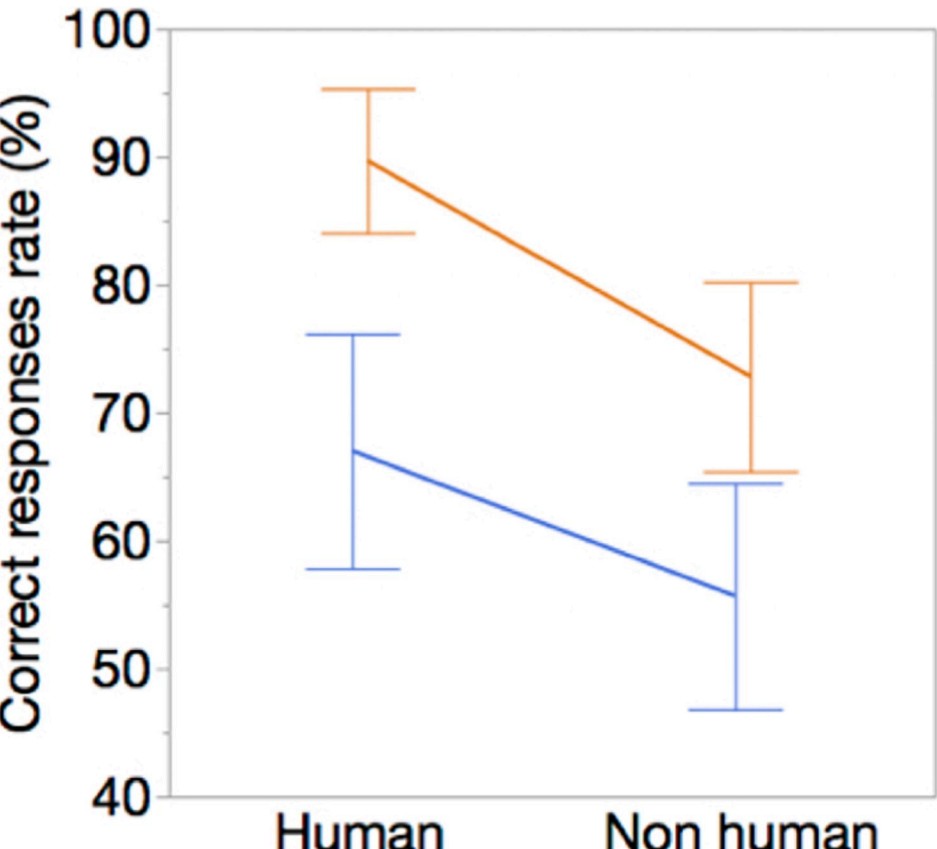

**Fig 3. Effects of Expertise and Attribution (human vs non-human marks) on the correct response rate.** Orange: Expert, blue: Controls. Error bars represents the confidence interval (95%).

the experts reported having used this information to attribute a European or African origin to the engravings (which was not asked of them). Some participants also reported paying attention to the support of the marks, the depth of the marks, and the number of marks. Despite a higher rate of correct responses for Experts than Controls, Expertise had no effect on the decision criteria reported by subjects in the debriefing ($p > .05$ for all chi-squared tests).

## Neuroimaging results

**Selection of hROIs.** The comparison of the Attribution and Orientation conditions evidenced 64 hROIs that were significantly more activated in Attribution than in Orientation (Fig 4, and see S2 and S3 Tables). They included the occipito-temporal regions, lateral occipital cortex, anterior insula, parahippocampal cortex, hippocampus, medial frontal cortex, anterior cingulate and at the subcortical level, thalamus and caudate nuclei. The effect of expertise and the type of judgement (*i.e.* human or non-human) were explored within this set of hROIs.

**Effect of Expertise and Attribution on BOLD activations in the 64 selected hROIs.** To assess whether Expertise interacts with Attribution and hROIs to modify BOLD levels, we set their 3-way interaction as fixed effects in a mixed-effect linear regression model. We observed no interaction between Expertise, Attribution, and hROIs ($F_{(63,1827)} = 0.63$, $p = 0.99$) nor between Expertise and Attribution $F_{(1,29)} = 0.01$, $p = 0.90$). This suggests that differences in brain region between attribution of human and non-human origin of the marks were the same in Experts and Controls.

Left                    Right

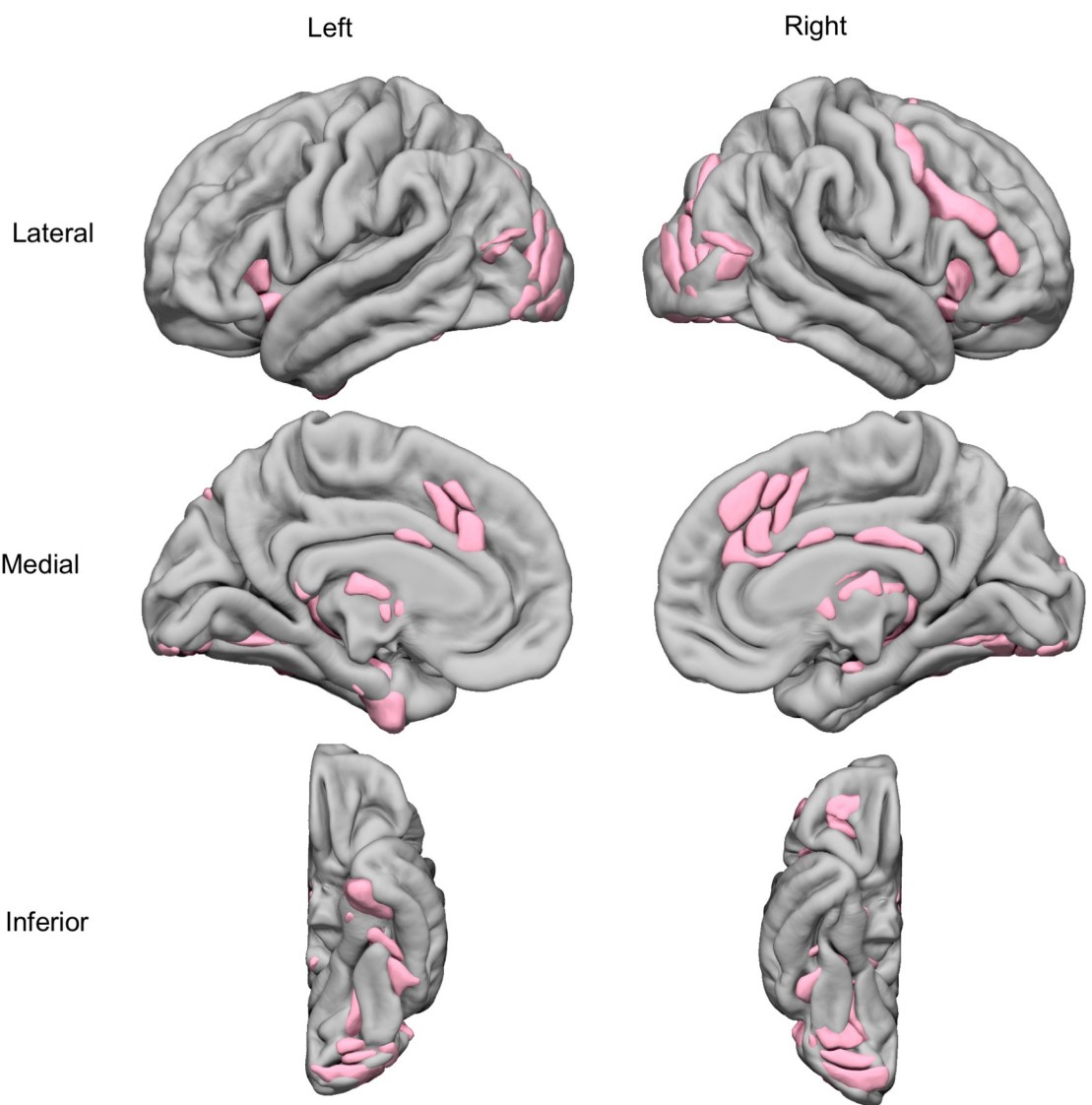

Lateral

Medial

Inferior

**Fig 4. Superimposition on an MRI template of the 64 hROIs activated during the [Attribution minus Baseline] condition and showing a significant BOLD signal increase in the Attribution minus Orientation contrast (p < 0.05, FDR corrected).**

**Effect of expertise.** We found that regional BOLD response differed between Experts and Controls (Expertise X hROI interaction: $F_{(63,1827)}$ = 2.14, p < .0001). *Posthoc* analysis revealed that visual areas were more activated by Experts than by Controls (Fig 5). It included regions belonging to the lateral occipital cortex, the occipital pole (all p < .05, FDR corrected) and a part of the left fusiform gyrus that nearly reached significance after correction for multiple testing (p = .02, uncorrected). In addition, Experts activated the anterior medial thalamus more strongly (p < .05, corrected), while a more posterior part of the medial thalamus did not survive correction (p = .04, uncorrected). No region was more activated in Controls than in Experts.

**Effect of attribution.** We found that regional BOLD response differed according to the type of judgment expressed during the Attribution condition (Attribution X hROI interaction: $F_{(63,1827)}$ = 2.87, p < .0001). Post-hoc analysis revealed that regions belonging to the anterior

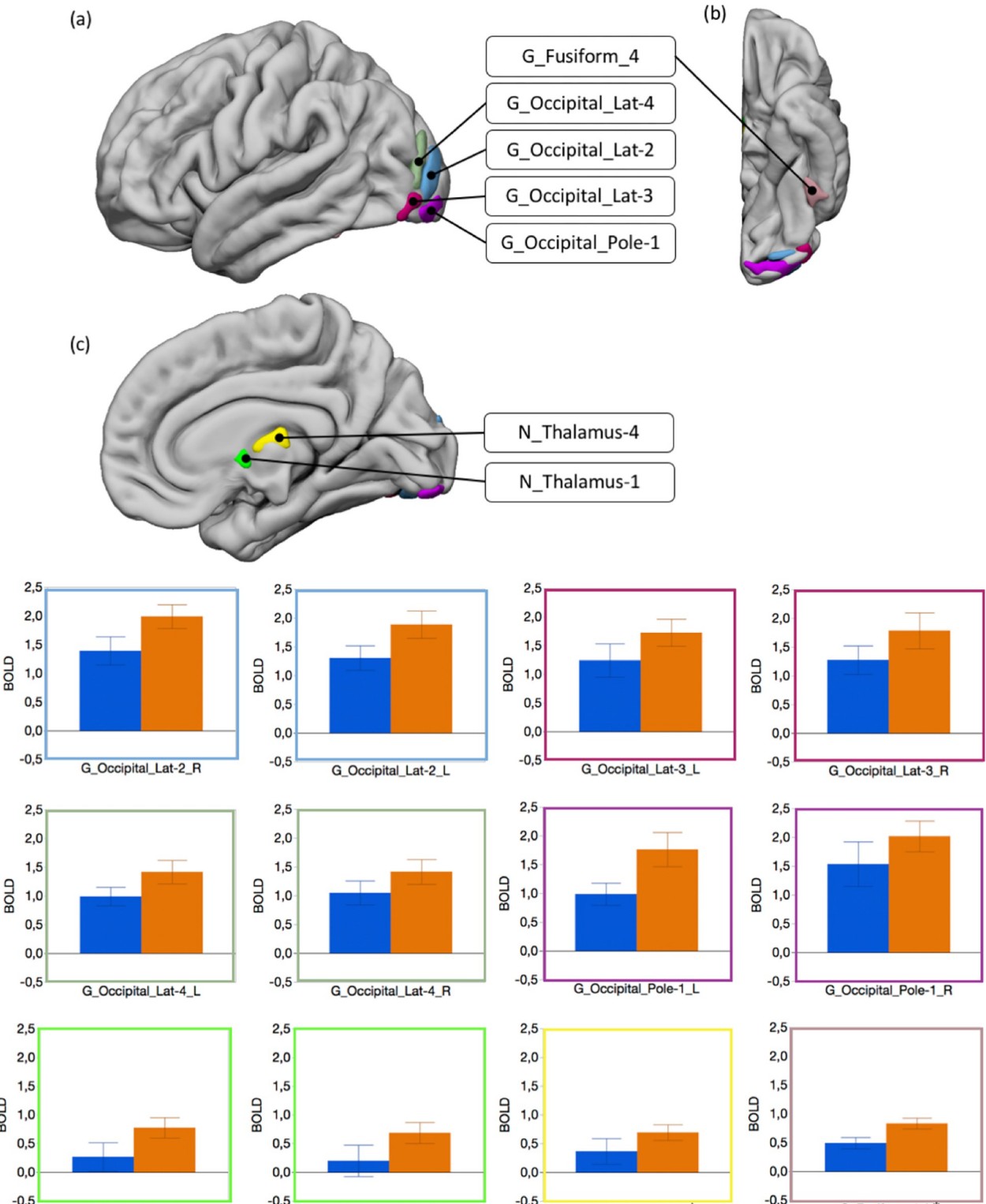

**Fig 5. Experts compared to controls in the judgment task.** Top: hROIs that showed a greater activity in Experts than in Controls. *: G_Fusiform-4_L and N_Thalamus-4_R were significant at uncorrected threshold only ($p_{uncorr}$ = 0.015 and puncorr = 0.019, respectively). (a) Lateral view of the left hemisphere. (b) Inferior view of the left hemisphere. (c) Medial view of the left hemisphere. Bottom: plots of the BOLD values in these regions in Controls (blue) and Experts (orange). Error bars represents the confidence interval (95%).

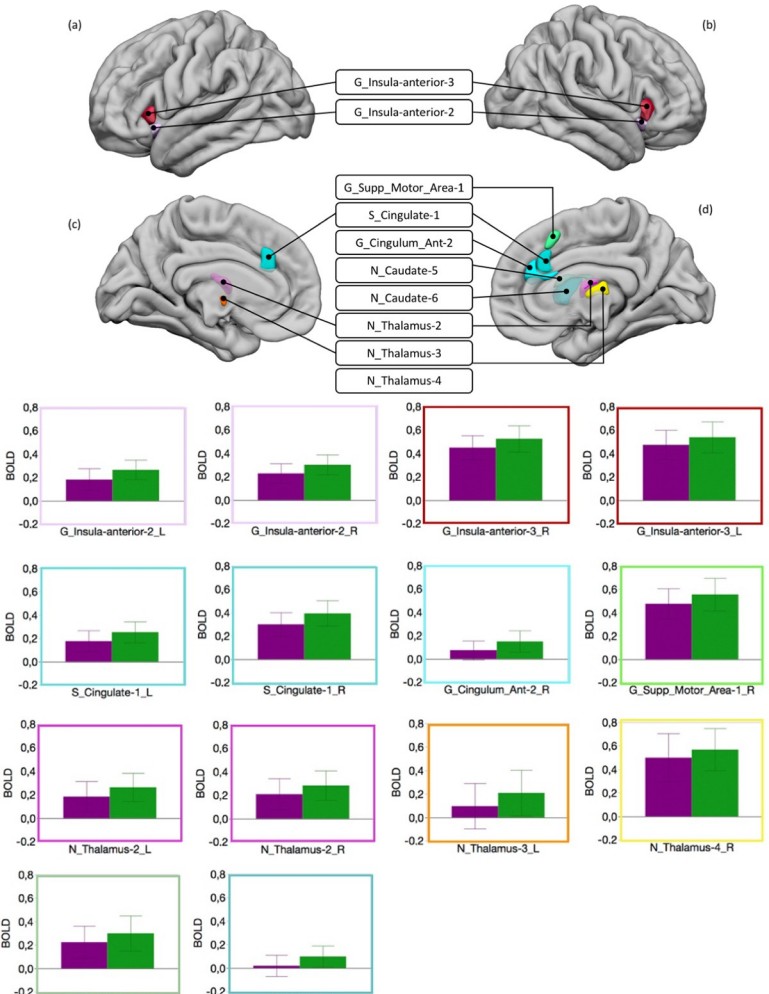

**Fig 6. Human *vs* non-human attribution.** Top: hROIs that showed a greater activity for non-human than for human attribution. (a) Lateral view of the left hemisphere. (b) Lateral view of the right hemisphere. (c) Medial view of the left hemisphere. (d) Medial view of the right hemisphere. Bottom: plots of the BOLD values in these regions for human attribution (purple) and non-human attribution (green). Error bars represents the confidence interval (95%).

insula, the anterior cingulate, the medial thalamus, and the right caudate nucleus were significantly more activated when a non-human origin was attributed to the marks (Fig 6, all p< .05, FDR corrected). No regions were more activated for the "Human" attribution.

## Discussion

This study aimed to characterise the effect of expertise in the perception of the earliest Palaeolithic abstract engravings at the behavioural and brain levels, using a judgment task between human-made engravings and surface modifications resulting from natural phenomena.

### Effect of expertise

During the Attribution condition of the judgement task, the participants had to decide whether the marks were intentionally human-made or the result of natural processes. This task was contrasted with an Orientation condition in which the same stimuli were used without

participants paying attention to the marks on the supports. Although the distinction criteria did not differ between experts and controls, the performances were significantly better for the experts. Note that archaeologists usually rely on much more refined analysis, not limited to a short visual analysis, to discern the human or natural origin of the marks. Nonetheless, the archaeologists confirmed their expertise in judging the natural or human origin of the engravings better than Controls, while they did not differ from them in the Orientation condition. As experts, the performances of archaeologists benefited from a greater ability to focus on the most discriminating elements, thus reducing the complexity of perceptual analysis. In addition, they could connect the perceptual analysis to knowledge stored in long-term memory and gained over many years and even decades. One could argue that these better performances reflected recognition of engravings previously encountered in the literature or their own research rather than an actual process of visual analysis. However, although a majority of experts recognised some of the engravings, only four recognised about ten, while the others recognised less than five. In addition, the experts were also better at identifying traces of natural origin for which a recollection process was unlikely, which supports the role of expertise in determining their higher performances. Finally, the brain regions more activated in the archaeologists than in the control participants do not correspond to the brain areas classically involved in long-term memory recall, such as the hippocampus, dorsolateral prefrontal cortex and parietal cortex [66–68].

During Attribution, Experts showed greater activation in the ventral part of the lateral occipital cortex and a strong trend in the left fusiform gyrus (G_Fusiform-4 in the AICHA atlas) in the occipito-temporal cortex (OTC). This result could reflect more discriminating visual analysis, which allowed a correct diagnosis of the origin of the marks. It has already been shown that the visual cortex and particularly OTC are involved in the visual processing of objects pertaining to the domain of expertise of the observer [69, 70]. For example, in a field that involves long-term acquired knowledge, as in the present study, it has been shown that experienced radiologists exhibit greater activation in OTC than less experienced ones when they detect lesions on chest radiographs [71, 72]. Most of the studies demonstrating the role of OTC in expertise have reported activation of a part of the fusiform gyrus called FFA [73–77]. It has been suggested that this region, which is crucial in face recognition, is more generally specialised in discriminating between stimuli that share common (prototypical) visual features and differences that are essentially accessible to the expert. This region is included in G_Fusiform-6 in the AICHA atlas and was not activated differently in Experts and Controls. Most of the studies that reported more activated FFA in experts relied on tasks favouring holistic processing (as in face recognition, [69]). In our study, participants based their decision on visual details (number of crossings, depth of marks) and were therefore processing the marks analytically rather than holistically. This could explain the lack of an expertise effect in this region, while it was present in adjacent areas.

The involvement of the "low level" visual areas was limited to a small region of the occipital pole (Fig 5, light purple blob), which was detected in both groups and more important in Experts than in Controls. Activity in the calcarine sulcus, which includes the primary visual area, did not increase during the attribution task compared to the Orientation task. This lack of activation argues against the hypothesis that low-level perceptual processes in this area are at the origin of the emergence of engravings production, as previously suggested [52, 56], even in subjects familiar with Palaeolithic marks. As a matter of fact, the vast majority of activations were in the associative visual cortex, including the OTC. The involvement of the visual cortex in this study illustrates its role in visual expertise. It does not fundamentally alter the conclusions of a previous study that highlighted the role of these regions in the visual analysis of

engravings [55]. In particular, it confirms that the visual analysis of the earliest abstract engravings engaged integrative visual areas involved in identifying visual percepts.

In the present work, Experts showed a greater involvement of the medial thalamus than Controls. The mediodorsal part of the thalamus is known to be involved in familiarity, corresponding to the impression that a percept or percepts of the same category have been experienced previously [60, 78]. In the present study, the archaeologists did not implement a different strategy from the control participants. Both groups relied on similar criteria to decide whether the engravings were of human or natural origin. The main difference is the long experience of archaeologists with both types of marks. Activation of the mediodorsal thalamus in the experts could reflect familiarity with these types of stimuli.

## Attributing a human or non-human origin to the marks

Our results showed that attributing a human or non-human origin to the marks is not equivalent, whether at the behavioural or the neural level. The lack of interaction between the Attribution, Expertise and hROIs indicated that the type of judgment (i.e. human or not human) did not affect BOLD differently in Experts and Controls. This is congruent with the absence of interaction between the attributed origin of the marks and the level of expertise at the behavioural level, indicating that both Experts and Controls made more errors for non-human than human attribution (with the Experts being better than controls in both categories). At the cerebral level, attributing a non-human origin to the marks resulted in greater activation in subcortical regions such as the head of the caudate nucleus and the thalamus and cortical areas including the anterior insula and the anterior cingulate, compared to assigning a human origin. All these regions belong to the so-called salience network [79–81]. This plays a fundamental role in detecting and selecting behaviourally relevant stimuli and is thus crucial in the decision-making process [82–84]. It is therefore not surprising that it was activated in our attribution task. The question is why it was activated more by the "non-human" choice than by the "human" choice. A meta-analysis showed that the activity in this network increased with uncertainty [85]. The rate of correct responses indicated that deciding that a mark was non-human was more uncertain than the opposite choice and might have triggered the greater activation of the salience network. This hypothesis is further supported by the fact that the anterior insula and anterior cingulate cortex would be particularly active during decision-making in a context of strong perceptual ambiguity [86, 87].

Interestingly, it has recently been shown that the cingulate and insular cortex in the salience network were involved in attributing others' intentions [88]. In addition, the anterior insula region is also generally associated with the sense of agency, *i.e.*, the awareness of who performs an action [89]. In the present study, the participants discriminated between marks resulting from human intention and those caused by fortuitous natural events. The processes associated with this choice likely contributed to the mobilisation of the cingulate and insular regions, thus suggesting that the salience network could be involved in attributing an origin to the outcome of an action, in addition to its role in attributing an action or intention. Notably, the regions concerned belong to the dorsal part of the salience network, mainly involved in cognition [90]. Interestingly, this subnetwork has not been found in the macaque, suggesting that it is engaged in human-specific abilities [91]. Distinguishing between human production and natural marks could be part of these functions.

## Conclusion

In a first study, we showed that the perception of schematic engravings engaged visual associative areas similar to those involved in object recognition [55]. This result was compatible with

a representational function of the engravings. The present study represents a further step. Whereas the first study was based on a brief presentation of schematised engravings, the experimental protocol of the present study involved a more careful inspection of actual pictures to recognize intentionally-made engravings from non-human marks. In addition, this study allowed the effect of expertise to be characterised, as well as the direct comparison of attributing human or not human origin to abstract marks. The comparison of activations between archaeologists and controls showed that the effect of familiarity mainly concerned visual associative areas, confirming their central role in the visual processing of engravings. The results showed that it was easier to correctly attribute a human than a non-human origin to the marks, whichever the expertise level, but that the nature of the attribution did not bear on visual regions. Since Palaeolithic abstract patterns resulted from human intention, the judgment concerning their attribution involved the salience network, which plays a pivotal role in perceptual decision-making and attribution of intention. The present study indicates that the visual processing of the earliest known engravings involves two categories of brain regions: 1. visual regions and, more specifically, associative visual areas for the processing of their global visual organisation, some of which are sensitive to familiarity, and 2. the salience network, which is necessary for deciding whether the marks result from a human intention. This result confirms that mere and exclusive processing of abstract engravings by the primary visual cortex is unlikely to explain their emergence and pristine perception, which required actions, intentions and the brain areas to infer the communicative potential of visual patterns.

## Supporting information

**S1 Table. Contextual and descriptive data on early engravings used as visual stimuli.**
(DOCX)

**S2 Table. Mean value and standard deviation of the BOLD signal in the 64 hROIs activated by at least one of the two groups of participants in Attribution minus Orientation contrast.**
(DOCX)

**S3 Table. MNI coordinates of the 64 hROIs activated by at least one of the two groups of participants in Attribution minus Orientation contrast.**
(DOCX)

## Acknowledgments

The authors thank Ginesis Lab (Labcom Programme 2016, ANR 16LCV2-0006-01) for their help in data management and processing. They are also indebted to Violaine Verrecchia for her help in data analysis.

## Author Contributions

**Conceptualization:** Francesco d'Errico, Emmanuel Mellet.

**Formal analysis:** Sandrine Cremona, Emmanuel Mellet.

**Funding acquisition:** Francesco d'Errico, Emmanuel Mellet.

**Investigation:** Sandrine Cremona, Emmanuel Mellet.

**Methodology:** Mathilde Salagnon, Sandrine Cremona, Francesco d'Errico, Emmanuel Mellet.

**Supervision:** Francesco d'Errico, Emmanuel Mellet.

**Visualization:** Mathilde Salagnon.

**Writing – original draft:** Mathilde Salagnon, Marc Joliot, Francesco d'Errico, Emmanuel Mellet.

**Writing – review & editing:** Sandrine Cremona.

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
