## [Decision Letter · Decision Letter 0]

26 Apr 2022

PONE-D-21-39820Neural correlates of perceiving and interpreting engraved prehistoric patterns as human production: effect of archaeological expertise.PLOS ONE

Dear Dr. Mellet,

Thank you for submitting your manuscript to PLOS ONE. After careful consideration, we feel that it has merit but does not fully meet PLOS ONE’s publication criteria as it currently stands. Therefore, we invite you to submit a revised version of the manuscript that addresses the points raised during the review process.

We look forward to receiving your revised manuscript.

Kind regards,

Enza Elena Spinapolice, Ph.D

Academic Editor

PLOS ONE

Journal Requirements:

This work was supported by the CNRS project 80 Prime Neurobeads, a grant from the IdEx Bordeaux/CNRS (PEPS 2015). Francesco d’Errico’s work is supported by the European Research Council through a Synergy Grant for the project Evolution of Cognitive Tools for Quantification (QUANTA), No. 951388; the Research Council of Norway through its Centres of Excellence funding scheme, SFF Centre for Early Sapiens Behaviour (SapienCE), project number 262618, the Talents Programme the Bordeaux University [grant number: 191022_001] and the Grand Programme de Recherche ‘Human Past’ of the Initiative d’Excellence (IdEx) of the Bordeaux University.

EM & FD: CNRS project 80 Prime Neurobeads, a grant from the IdEx Bordeaux/CNRS (PEPS 2015).

FD: European Research Council through a Synergy Grant for the project Evolution of Cognitive Tools for Quantification (QUANTA), No. 951388; the Research Council of Norway through its Centres of Excellence funding scheme, SFF Centre for Early Sapiens Behaviour (SapienCE), project number 262618, the Talents Programme the Bordeaux University [grant number: 191022_001] and the Grand Programme de Recherche ‘Human Past’ of the Initiative d’Excellence (IdEx) of the Bordeaux University.

Reviewers' comments:

Reviewer's Responses to Questions

**Comments to the Author**

1. Is the manuscript technically sound, and do the data support the conclusions?

Reviewer #1: Partly

Reviewer #2: Partly

2. Has the statistical analysis been performed appropriately and rigorously? 

Reviewer #1: Yes

Reviewer #2: Yes

3. Have the authors made all data underlying the findings in their manuscript fully available?

Reviewer #1: Yes

Reviewer #2: No

4. Is the manuscript presented in an intelligible fashion and written in standard English?

Reviewer #1: Yes

Reviewer #2: Yes

5. Review Comments to the Author

Reviewer #1: This is a welcome article which, if only to an extent, puts to rest the debate concerning the neural perception of engraved prehistoric marks. It mainly takes its cue from the 2019 article by some of the same authors, ‘Neuroimaging supports the representational nature of the earliest human engravings’, R Soc Open Sci. Indeed, the most important claim and conclusion of this article rests in rebuking the involvement of the primary visual cortex or low-level visual areas in the perception of the engravings.

I have some misgivings, however, concerning how the experiment was carried out. The experiment involved expert and non-expert participants who were asked to recognize human vs natural (non-human made) engravings on 21 prehistoric archaeological objects.

Here a few reservations and questions, as follows:

1. How was the sample of the prehistoric objects selected? It is not clear, at least to me, what criteria were used.

2. Number of participants to the experiment are low. Of course, out of 15-16 participants the margins of error will skew the overall results substantially: all it takes is one participant to the experiment to get an answer wrong and the percentage results plummet.

3. I feel it should be stressed how including experts can be an advantage in reaching neutral results. The authors claim that they wanted to estimate the effect of familiarity and prior knowledge. But I see no empirical advantage in doing this (and even despite this, it is surprising that only 81% of the expert participants gave correct answers, but see above 2). This feels very circular to me, even when it comes to the main research question: of course, experts will recognize the human-made marks and will have a neural activation that involves the visual areas in the OTC, the occipital pole and part of the left fusiform gyrus. In other words, it is expected that the effect of expertise will be felt in the perception of abstract marks, whether low-level visual areas are involved or not.

4. Contrasting experts vs non-experts feels, therefore, inconsequential to the research question. If advanced expertise will (and does) condition the results, a contrast with results from non-experts will lead to clearly biased conclusions or a warped dichotomy, the very half of which is patently expected.

5. The two questions that the authors asked the participants are Attribution (human-made or not) and Orientation (‘is the longest axis of the medium on which the marks are present vertical?’). This latter point is not explained, nor is its importance stated clearly- what role plays verticality? This should be clarified. Partially tied to this are the criteria upon which the participants based their decisions in the debriefing session: how do marks relate to the orientation?

Reviewer #2: The authors present an analysis about the neural correlates of perception, recognition and interpretation of Palaeolithic engravings compared with patterns of unintentional origin by experts (archaeologists) and non-experts to assess the communicative potential of the engravings. Analyses carried out with a rigorous protocol show that there is a clear ability by both experts and non-experts to discriminate between patterns of natural and intentional origin. Interestingly, the analysis highlights the activation in the perception and interpretation of archaeological patterns of brain areas afferent to the ventral pathway of the brain related to the integration and semantic interpretation of visual data. While the primary visual cortex is involved at limited extent. This is an important result because it allows to exclude that the engravings are the expression of kinaesthetic dynamics as proposed by previous authors. The analyses show a difference between experts and non-experts in the brain areas involved in the perception and interpretation of Palaeolithic engravings compared to natural patterns. In particular, in interpreting patterns as natural, archaeologists show a greater involvement of the so-called salience network involved in decision making processes. This is interpreted by the authors as an effect of familiarity and awareness in performing actions. It is interesting (as counterintuitive) that this network is activated in experts more in the attribution of natural rather than intentional activities. However, this result lends itself to some criticism that the authors have not duly justified in the current version of the article.

In particular, there is an unresolved bias related to the fact that the experts' response may be linked not to familiarity with the actions required to create meaning-bearing engravings but to mere prior knowledge of them. In fact, in the analyses were used Palaeolithic engravings, which were well known to archaeologists. It cannot therefore be excluded that this initial recognition (or non-recognition in the case of natural patterns) strongly conditioned their choice and the neurophysiological response. The greater activation of the fusiform gyrus in the archaeologists would suggest a holistic recognition (as reported by the same authors) of the patterns in which the influence of a previous knowledge of the same cannot be excluded. Even the very high percentage of correct attributions by the archaeologists suggests that phenomena of prior knowledge interfere with the results. The low level of negative responses may be linked to the small number of expert individuals tested (n=15). From this point of view, it would be advisable (if possible) to increase the sample size of expert and non-expert groups.

Other criticisms include the fact that the authors use Palaeolithic engravings of different chronological ages and geographical origins without providing adequate information about them. The engravings shown result in a heterogeneous group referring to different human populations with possible differences in cognitive abilities. In Europe, the engravings are mainly parallel in pattern, whereas African engravings often show cross-patterns; this information should be discussed when discussing the relevance (or otherwise) of the orientation in the tasks of recognition of the engravings.

In summary, although interesting and methodologically rigorous, this study cannot be published until the expert group bias has been resolved. The authors should provide valid and proven arguments to exclude the possibility that other cognitive processes related to the prior knowledge about the carvings shown could make the results so ambiguous and biased in their interpretation.

6. PLOS authors have the option to publish the peer review history of their article (what does this mean?). If published, this will include your full peer review and any attached files.

Reviewer #1: No

Reviewer #2: No

---

## [Author Response · Author response to Decision Letter 0]

24 May 2022

# Reviewer 1

Question: How was the sample of the prehistoric objects selected? It is not clear, at least to me, what criteria were used.

Response: We better explain our criteria in the revised version of the manuscript and have added a table in the Supporting Information providing information on each of the objects. 

Line 168: “The 21 pictures of engravings included in the study were abstract engravings, dated between 800 ka to 30 ka, not found in Upper Paleolithic contexts in association with figurative art, have demonstrated anthropogenic origin (29,59,60), and were recognizable on a photo of the object on which they occur. The number of items (21 human and 21 non-human) was chosen in order not to tire the participants since we adopted a so-called slow event-related paradigm (a 3s presentation every 9.5s). The engravings come from African and Eurasian sites, and are attributed to Homo erectus, Neanderthals and Early Modern Humans”

Question: Number of participants to the experiment are low. Of course, out of 15-16 participants, the margins of error will skew the overall results substantially: all it takes is one participant to the experiment to get an answer wrong and the percentage results plummet.

Response: We agree that the small number of subjects may suggest at a first sight that the results are not sufficiently reliable. However, our sample did not include only 16 participants. It was composed of 16 Experts and 15 Controls, i.e. 31 participants in total. If the variability of the experts’ responses would have been too large, the results would have not been statistically significant. We observe the opposite. We performed again the analyses by attributing random answers (50% correct answers for both types of traces) to the participant who obtained the best scores in the discrimination task (100% recognition of traces of human origin, 90% identification of traces of natural origin). The results remained highly significant (p=0.0002). Note that the power calculation to detect a 20% difference in correct responses between our two groups (31 participants in total) is .89. We conclude that wrong answers from an expert participant would have not been able to change the outcome of the experiments.

Question: I feel it should be stressed how including experts can be an advantage in reaching neutral results. The authors claim that they wanted to estimate the effect of familiarity and prior knowledge. But I see no empirical advantage in doing this (and even despite this, it is surprising that only 81% of the expert participants gave correct answers, but see above 2). This feels very circular to me, even when it comes to the main research question: of course, experts will recognize the human-made marks and will have a neural activation that involves the visual areas in the OTC, the occipital pole and part of the left fusiform gyrus. In other words, it is expected that the effect of expertise will be felt in the perception of abstract marks, whether low-level visual areas are involved or not.

Response: We are not sure what the reviewer means by neutral results in this context and have troubles to understand what the precise target of his/her remark is. However, it seems that the study’s objectives were not presented clearly enough. We have addressed this issue in the revised version of the manuscript by adding a sentence (see below). 

In previous work, we showed that the perception of Paleolithic engravings by subjects with no expertise in Palaeolithic archaeology involved the occipitotemporal cortex (OTC). However, we can reasonably assume that our ancestors were familiar with the engravings they produced. To facilitate drawing inferences on the brain functioning of past humans it was therefore necessary to include participants whose knowledge approached this familiarity. This is achieved in the manuscript submitted to PLOS ONE, in which we compare a group of subjects with prior knowledge of engravings to a group of subjects without such knowledge (but matched on the criteria of age, gender and education level), in which we contrast differences in brain activations between the two groups with and without such prior knowledge. The statement by the reviewer according to which 81% is a surprisingly low score for experts need in our view to be nuanced. Discriminating natural from actual engravings can be challenging in some cases even for experts and one has to consider the unusual conditions in which the experiment take place (lying in the fMRI). Thus, an average of 81% confirms rather than infirms the experts’ ability to identify archaeological engravings.

In order to clarify the aims of the study we added the following paragraph in the introduction (Line 100):

“In our previous study (55), the participants lacked archaeological knowledge. The brain regions mobilised by the perception of the engravings be altered according to the level of familiarity that the subjects have with these productions. The inclusion of participants with this familiarity allows approaching the knowledge that the engravers probably possessed and thus avoid a novelty effect at the brain level in the participants (59,60).”

Question: Contrasting experts vs non-experts feels, therefore, inconsequential to the research question. If advanced expertise will (and does) condition the results, a contrast with results from non-experts will lead to clearly biased conclusions or a warped dichotomy, the very half of which is patently expected.

Response: Contrasting the response of populations with different degrees of expertise is a common practice in fMRI studies and is considered a reliable way to explore differences in neural networks involved in cognitive tasks (Gauthier et al, 1999; Dehaene et al, 2010, Wan et al, 2011; Bilalić et al, 2012, Wang et al, 2020). This approach has also been successfully applied in studies devoted to infer trends in past cognitive evolution (Stout et al., 2008). We think that in our study, as in those cited here above, contrasting the response of experts and non-experts provides information directly related to the research question. The goal in our study was to characterise the brain regions involved in differentiating human-made from natural markings. Such hypothesis testing approach may have led to results indicating that the same or different brain regions were involved. Therefore, we have difficulties to share this reviewer's opinion that this approach would have necessarily led to “clearly biased conclusions or a warped dichotomy, the very half of which is patently expected”. Our results are presented in Figure 4 of the article, showing that during the assigned task, both experts and non-experts mobilised common brain regions, including the OTC, thus confirming results from a previous study (although the perceptual task was very different). In other words, comparing a group of subjects with knowledge about the engravings and naive subjects allowed us to highlight the effects of this knowledge on the activation of the regions concerned by the attribution task.

Question: The two questions that the authors asked the participants are Attribution (human-made or not) and Orientation (‘is the longest axis of the medium on which the marks are present vertical?’). This latter point is not explained, nor is its importance stated clearly- what role plays verticality? This should be clarified. Partially tied to this are the criteria upon which the participants based their decisions in the debriefing session: how do marks relate to the orientation?

Response: We clarify the role of this task in the revised draft. The orientation task is a control condition. It used the same images as the attribution condition in a task that does not require visual analysis of the marks (defining the orientation of the longest axis of the object without paying attention to the marks present on them). When we subtract the activations of the orientation task from those of the attribution task, we eliminate all the activations that are not specific to the latter (low-level perceptual processes such as contrast, luminance, perception of the shape of the support etc. but also motor activity related to button press). This task did not require any archaeological knowledge, as confirmed by the lack of difference in the number of correct responses between the experts and the controls in this condition.

The paragraph now reads in the revised version of the manuscript (Line 150): “The orientation task is a control condition. It used the same images as the attribution condition in a task that does not require visual analysis of the marks (defining the orientation of the longest axis of the object without paying attention to the marks present on them) nor any archaeological knowledge. When subtracting the activations of the orientation task from those of the attribution task, all the activations that are not specific to the latter (low-level perceptual processes such as contrast, luminance, perception of the shape of the support, motor activity related to button press…) are cancelled-out.”

# Reviewer 2

Question: In particular, there is an unresolved bias related to the fact that the experts' response may be linked not to familiarity with the actions required to create meaning-bearing engravings but to mere prior knowledge of them. In fact, in the analyses were used Palaeolithic engravings, which were well known to archaeologists. It cannot therefore be excluded that this initial recognition (or non-recognition in the case of natural patterns) strongly conditioned their choice and the neurophysiological response.

Response: The reviewer is right. A part of the correct responses given by the experts is due to the fact that they recognized engravings that they encountered before. However, recognition of a known item (also known as recollection, Tulving, 1985; Gardiner & Richardson-Klavehn, 2000) cannot, in our opinion, explain the results.

• First, no expert indicated during the debriefing following the experiments that they recognized all the engravings presented to them. Four participants reported that they recognized ten out of the 21 engravings presented. The other experts recognized less than five. This can be due to 1) the fact that the experts were archaeologists with expertise in Palaeolithic art and the techniques used in Palaeolithic times to produce graphic expressions but not necessarily in the earliest abstract engravings; 2) the viewing conditions were different from the observation conditions archaeologist are used to; 3) in some cases only a part of the engraving was presented. In addition, the experts reported during the debriefing that they focused on the form and organization of the markings, the same answer given by the naïve participants, and not recognizing previously seen markings.

• Second, Experts performed significantly better than Controls not only in the attribution of human marks but also in the identification of natural marks. Their high rate of correct responses for natural items cannot be explained by a recognition process. It clearly reflects the implementation of expertise acquired during years of observations, their higher ability of categorization being based on their experience in this material.

• Third, the brain regions that were more activated in experts than in controls were visual integrative regions (OTC) and subcortical regions that are not involved in the retrieval from long-term memory of an item. In particular, there was no difference between experts and controls in the hippocampal (Bird, 2017), dorsolateral prefrontal or parietal regions known to be central to these memory processes. In particular, the hippocampus is involved in recollection memory while not impacted by familiarity (Montaldi et al., 2006). The lack of difference between the two groups in these areas does not support the hypothesis that known item identification processes were predominant in the experts during the task.

The corresponding paragraph (line 356) now reads: “One could argue that these better performances reflected recognition of engravings previously encountered in the literature or their own research rather than an actual process of visual analysis. However, although a majority of experts recognised some of the engravings, only four recognised about ten, while the others recognised less than five. In addition, the experts were also better at identifying traces of natural origin for which a recollection process was unlikely, which supports the role of expertise in determining their higher performances. Finally, the brain regions more activated in the archaeologists than in the control participants do not correspond to the brain areas classically involved in long-term memory recall, such as the hippocampus, dorsolateral prefrontal cortex and parietal cortex (66-68 ).”

Question: The greater activation of the fusiform gyrus in the archaeologists would suggest a holistic recognition (as reported by the same authors) of the patterns in which the influence of a previous knowledge of the same cannot be excluded.

Response: We thank the reviewer for this suggestion. However, we think that adopting a holistic analysis is not necessarily related to a long-term memory recall of an item but may correspond to an analysis of its global spatial organisation.

In fact, it is difficult to know whether the participants applied a local or global strategy. According to their verbal report, experts and controls appear to have been looking at the details of the marks, thus adopting a more local than holistic visual analysis. The part of the fusiform gyrus that would respond to holistic processes is the Fusiform Face Area (FFA), for which, as we have written, we found no difference between experts and controls. So even if a holistic strategy was privileged, it was used by both groups, which does not support the reviewer's suggestion of a recognition-related activation. Finally, it has been shown that high spatial frequency, such as the local organisation of the engraved patterns, is predominately processed by the visual cortex of the left hemisphere (Peyrin et al., 2014), as was the activation we reported in the fusiform gyrus.

Question: Even the very high percentage of correct attributions by the archaeologists suggests that phenomena of prior knowledge interfere with the results.

Response: We agree with the reviewer that prior knowledge played a role (see our comments above). However, prior knowledge also includes the archeologists’ ability to identify markings left by humans in the past even in cases in which the specific engraving was unknown to them. This is the so-called “visual expertise”. It refers to archaeologists recognising the type of marks left by humans without necessarily knowing them individually. The same process allows them to recognise better than non-archaeologists the traces produced by natural processes or carnivores.

Question: The low level of negative responses may be linked to the small number of expert individuals tested (n=15). From this point of view, it would be advisable (if possible) to increase the sample size of expert and non-expert groups.

Response: We agree that the small number of subjects may suggest at a first sight that the results are not sufficiently reliable. However, our sample did not include only 16 participants. It was composed of 16 Experts and 15 Controls, i.e. 31 participants in total. If the variability of the experts’ responses would have been too large, the results would have not been statistically significant. We observe the opposite. We would have loved to include more archaeologists experts in Palaeolithic engravings. Fifteen experts in this rather specialized field is already a relatively large number and, of course, not all experts were available.

Question: Other criticisms include the fact that the authors use Palaeolithic engravings of different chronological ages and geographical origins without providing adequate information about them.

Response: We have added in the Supporting Information a table (S1 Table) providing information on each engraving used in the experiment. We better explain our selection criteria in the revised version of the manuscript. “The 21 pictures of engravings included in the study were abstract engravings, dated between 800 ka to 30 ka, not found in Upper Paleolithic contexts in association with figurative art, have demonstrated anthropogenic origin (29,59,60, see S1 Table), and were recognizable on a photo of the object on which they occur. The number of items (21 human and 21 non-human) was chosen in order not to tire the participants since we adopted a so-called slow event-related paradigm (a 3s presentation every 9.5s). The engravings come from African and Eurasian sites, and are attributed to Homo erectus, Neanderthals and Early Modern Humans”.

Question: The engravings shown result in a heterogeneous group referring to different human populations with possible differences in cognitive abilities. 

Response:

See above for our selection criteria. Our study did not aim to investigate potential cognitive differences between fossil human species or geographic areas. There are not enough engravings to do that. We assume that the ability to attribute a human origin to engraved patterns must have been common to hominins populations able to produce engravings.

Question: In Europe, the engravings are mainly parallel in pattern, whereas African engravings often show cross-patterns; this information should be discussed when discussing the relevance (or otherwise) of the orientation in the tasks of recognition of the engravings.

Response: 

As reported in the debriefing results, most of the participants took into account the presence of parallel or crossed patterns, generally to attribute a human origin to the engravings. However, none of the experts reported having used this information to attribute a European or African origin to the engravings (which was not asked of them).

Interestingly, some experts have pointed out that these patterns could be misleading in attributing a human origin to the marks since parallel and crossed lines could also be found in the natural marks.

The following sentence has been added in the revised manuscript Line 265: “Note that the engravings of European origin are mainly made of parallel in pattern, whereas African engravings often show cross-patterns However, none of the experts reported having used this information to attribute a European or African origin to the engravings (which was not asked of them).”

Question: In summary, although interesting and methodologically rigorous, this study cannot be published until the expert group bias has been resolved. The authors should provide valid and proven arguments to exclude the possibility that other cognitive processes related to the prior knowledge about the carvings shown could make the results so ambiguous and biased in their interpretation.

Response: We hope that the clarifications expressed in our responses, the changes introduced in the main text and data added in Supplementary Information will convince this reviewer that the paper is worth publishing. We think we have provided supplementary evidence indicating that the experts' decision was not or only negligibly determined by recognition of already perceived items. We briefly recall such evidence here:

1. None of the participants reported having recognised all the engravings (Four experts recognised half of the engraving, in the others cases no expert recognized more than four).

2. The experts’ best performances also concerned marks of natural origin (for which recollection is very unlikely).

3. The brain regions more activated in the archaeologists than in the control participants do not correspond to the regions classically involved in long-term memory recall.

---

## [Decision Letter · Decision Letter 1]

7 Jul 2022

Neural correlates of perceiving and interpreting engraved prehistoric patterns as human production: effect of archaeological expertise.

PONE-D-21-39820R1

Dear Dr. Mellet,

We’re pleased to inform you that your manuscript has been judged scientifically suitable for publication and will be formally accepted for publication once it meets all outstanding technical requirements.

Kind regards,

Enza Elena Spinapolice, Ph.D

Academic Editor

PLOS ONE

Additional Editor Comments (optional):

Reviewers' comments:

Reviewer's Responses to Questions

**Comments to the Author**

1. If the authors have adequately addressed your comments raised in a previous round of review and you feel that this manuscript is now acceptable for publication, you may indicate that here to bypass the “Comments to the Author” section, enter your conflict of interest statement in the “Confidential to Editor” section, and submit your "Accept" recommendation.

Reviewer #1: All comments have been addressed

Reviewer #2: All comments have been addressed

2. Is the manuscript technically sound, and do the data support the conclusions?

Reviewer #1: Partly

Reviewer #2: Yes

3. Has the statistical analysis been performed appropriately and rigorously? 

Reviewer #1: Yes

Reviewer #2: Yes

4. Have the authors made all data underlying the findings in their manuscript fully available?

Reviewer #1: Yes

Reviewer #2: Yes

5. Is the manuscript presented in an intelligible fashion and written in standard English?

Reviewer #1: Yes

Reviewer #2: Yes

6. Review Comments to the Author

Reviewer #1: I am satisfied with the answers offered on points of detail, although I am not completely convinced by the method employed in relation to experts.

Reviewer #2: The authors responded positively to all my comments. The paper of interest to the PlosOne audience can be accepted. Thank you

7. PLOS authors have the option to publish the peer review history of their article (what does this mean?). If published, this will include your full peer review and any attached files.

Reviewer #1: No

Reviewer #2: No

---

## [Editor Report · Acceptance letter]

12 Jul 2022

PONE-D-21-39820R1 

Neural correlates of perceiving and interpreting engraved prehistoric patterns as human production: effect of archaeological expertise 

Dear Dr. Mellet:

I'm pleased to inform you that your manuscript has been deemed suitable for publication in PLOS ONE. Congratulations! Your manuscript is now with our production department. 

Kind regards, 

on behalf of

Dr. Enza Elena Spinapolice 

Academic Editor

PLOS ONE